# Ophthalmic Ultrasonography in Sub-Saharan Africa—A Kinshasa Experience

**DOI:** 10.3390/diagnostics11112009

**Published:** 2021-10-28

**Authors:** Georgette Ngweme, M.T. Ngoyi Bambi, Longo Flavien Lutete, Ngoy Janvier Kilangalanga, Adrian Hopkins, Oliver Stachs, Rudolf Friedrich Guthoff, Thomas Stahnke

**Affiliations:** 1Centre de Formation Ophtalmologique Pour l’Afrique Centrale, Eye Department, Saint Joseph’s Hospital, Kinshasa P.O. Box 322, Congo; ngwemegeorgette@yahoo.fr (G.N.); bambidiavita@yahoo.fr (M.N.B.); lutetelongo@gmail.com (L.F.L.); kilangalanga@yahoo.fr (N.J.K.); 2Programme National de Santé Oculaire et Vision (PNSOV), Kinshasa P.O. Box 322, Congo; adrianhopkinsconsulting@gmail.com; 3Department of Ophthalmology, Rostock University Medical Center, 18057 Rostock, Germany; oliver.stachs@med.uni-rostock.de (O.S.); rudolf.guthoff@med.uni-rostock.de (R.F.G.)

**Keywords:** B-scan ultrasound, ultrasonography, dense cataract, posterior segment abnormalities, ocular adnexal pathology, Sub-Saharan Africa

## Abstract

The aim of this study was to analyze the use of the diagnostic B-scan ultrasound. Should it be made accessible to all surgical centers in Sub-Saharan Africa in order to (i) avoid unnecessary cataract surgery and (ii) evaluate extraocular pathology? This study was conducted in Kinshasa from 2006 to 2019. Three hundred and twenty-three patients were included and separated into two groups. Group 1 included 262 patients with dense cataract. Group 2 consisted of 61 patients with pathologies of the ocular adnexa, and all were examined with a B-scan ultrasound. In group 1, there were 437 systematically screened eyes. Three hundred and ninety-eight eyes (91.08%) showed no abnormalities, 13 (2.97%) retinal detachments were identified, and 15 (3.43%) demonstrated a detached posterior hyaloid membrane. In the second group, 61 patients were examined (group 2). In 20 of them, surgery was performed for biopsy, tumor excision, mucoceles drainage, and palliative treatment. The need for routine B-scan examinations in dense cataract patients seems to be limited and can most likely be replaced by a thorough application of locally available examination techniques. B-scan application is recommended to manage orbital patients in the most cost-effective way.

## 1. Introduction

Diagnostic ultrasound was introduced in ophthalmology in the early 1950s [1] and was first applied prior to vitrectomy in the 1970s [2,3]. A-scan ultrasound for axial length measurements in cataract patients for individualized lens implantation was established and is still a recommended and mostly accepted standard procedure in ophthalmic practice in low- and middle-income (LMIC) countries [4,5]. B-scan ultrasound is a two-dimensional imaging system that can be used for evaluating abnormalities of the posterior segment of the eyes. It is particularly useful when the fundus is not accessible by a direct or indirect ophthalmoscopy evaluation, e.g., in the presence of dense cataracts. It is performed routinely in dense cataract as a preoperative examination for evaluating posterior segment abnormalities that may influence the visual prognosis after surgery. However, in LMIC countries, many patients do not have access to such procedures for socioeconomic reasons. Moreover, such sophisticated tests may not be available in many centers [6,7]. In addition, orbital pathology is poorly managed in many LMIC countries, and even in referral centers, sub-specialization in the orbit is rare. General ophthalmologists are reluctant to intervene and often prescribe costly scanning before considering the case.

The purpose of this study was to investigate whether, in resource poor countries, diagnostic B-scan ultrasounds should be made accessible to all surgical centers in Sub-Saharan Africa in order to (i) avoid unnecessary cataract surgery in eyes with untreatable posterior segment pathology and (ii) evaluate extraocular pathology, mainly orbital space occupying lesions, in particular, where computed tomography (CT) or magnetic resonance imaging (MRI) are not available.

## 2. Materials and Methods

The A- and B-scan diagnostic ultrasound system (OTI Inc., Toronto, ON, Canada) was made available to the Saint Joseph’s Hospital Eye Department in Kinshasa (DR Congo) by a Rotary global grant in 2006. While the A-scan was routinely used for biometry, the B-scan system was rarely used in routine diagnostics, and the equipment was often unusable due to low maintenance. The device was replaced by a further donation in 2018 by a laptop-based B-scan ultrasound system. Regular teaching lessons were held during annual visits of the Rostock group for both intraocular and ocular adnexal applications.

### Patients

Group 1: The retrospective cross-sectional study was conducted in the Department of Ophthalmology at Saint Joseph Hospital from August 2014 to November 2019. All patients with dense cataract underwent ultrasound examination by a B-scan as a preoperative examination. All patients similarly underwent a basic clinical examination, including visual acuity, pupillary reaction, light projection, and tonometry. The etiology of the cataracts was determined by patient history and examination and classified as (I) congenital, (II) juvenile, (III) diabetic, (IV) age-related, (V) posttraumatic, (VI) post-uveitic, or (VII) undetermined. The morphology and density of the cataract were determined clinically by a slit lamp examination. Only patients in whom the posterior segment was inaccessible during the clinical examination by ophthalmoscopy were included in the study. Patients younger than one year, patients already known to have posterior segment lesions, and those who had a previous history of ocular surgery were excluded from the study.

For patients suffering from significant posterior segment pathologies and retinal detachments, the clinical parameters were analyzed retrospectively. These parameters included posterior synechiae, ocular hypotony, and an inaccurate light projection test.

After completing the clinical examination, patients were evaluated using the A- and B-scan diagnostic ultrasound system (OTI Inc., Toronto, ON, Canada), replaced in April 2018 by the Master-Vu B-scan with a 12-MHz probe (Model No: MV5600, Sonomed Inc., Lake Success, NY, USA) coupled with methylcellulose on the lid surface. In each case, the examinations were performed by one of the first authors to guarantee standardized procedures and to avoid different diagnostic classifications.

The data collected during the study were recorded with MS Excel (Microsoft Inc., Redmond, WA, USA). None of the patients’ names were linked to the data. EPI Info™ 7 (Centers for Disease Control and Prevention, CDC) was used for statistical analysis; interpreting; and data presentation (mean, median, frequency, and percentage).

Group 2: A list of patients for review was established from the patients seen at St Joseph’s Hospital, as well as patients referred from other ophthalmologists in the city. These patients, seen together with members of the Rostock group for diagnostic assessment, including B-scan ultrasound examinations, in the years 2006–November 2019, were included in this group. This group was also taught. Suggestions for further patient treatment were offered, and in selected cases, surgical treatment was performed on-site. During this period, 61 patients with extraocular pathology were examined. Most patients were referred to Saint Joseph’s Hospital during the study period and were recalled when the Rostock group came to their next visit.

### Ethical Considerations

This study complied with local laws and the principles of the Declaration of Helsinki. Ethical approval was obtained from the Archidiocese de Kinshasa, Comité Scientifique, Saint Joseph’s Hospital. A consent form was presented, explained to patients, and their signature was required. Patients could stop participating in the study at any time without giving reasons.

## 3. Results

### 3.1. Intraocular Pathology

Between August 2014 and November 2019, 437 eyes from 262 cataract patients with dense cataracts were screened, including with a B-scan examination (52.19% female, 47.81% male; 66.79% of patients suffered bilateral cataract blindness). The age range was 1 year to 96 years, with a mean age of 52 years.

One hundred and seventy-five patients had bilateral cataract, and 87 were only the eyes. Two hundred and eighty-four (64.99%) eyes had age-related cataract, 92 (21.05%) eyes had congenital cataract, 36 (8.24%) eyes had diabetic cataract, 10 (2.29%) eyes had posttraumatic cataract, 8 (1.83%) eyes were diagnosed with post-uveitis cataracts, 4 (0.92%) eyes had undetermined cataract, and 3 (0.69%) eyes had juvenile cataract.

In B-scan ultrasonography, 39 (8.92%) eyes had findings suggestive of posterior segment abnormalities. The most common abnormality was posterior vitreous detachment in 15 (3.43%) eyes, followed by retinal detachment in 13 (2.97%) eyes. The findings are listed in Table 1. Three hundred and ninety-eight (91.1%) eyes demonstrated no posterior segment pathology in the B-scan ultrasonography.

Examples of findings of the B-scan ultrasonic examination of two eyes from different patients are demonstrated in Figure 1. One of the most frequent posterior abnormalities (retinal detachment) is shown in Figure 1a, and a rare finding (intraocular foreign body) is shown in Figure 1b.

Twenty-five patients (64.28%) presenting posterior segment lesions were in the age range between 40 and 69 years, with a peak at the range of 60–69 years (33.33%). None of the patients younger than 10 presented posterior segment lesions.

For the 13 patients with retinal detachment, the clinical findings were evaluated retrospectively. Posterior synechiae were found in 8 patients (61.54%), ocular hypotony in 4 (30.77%) patients, and inaccurate light projection in 10 patients (76.92%). Three patients did not show any of these clinical signs. Table 3 presents the frequency of the clinical parameters considered as risk factors for retinal detachment.

There was a total number of 13 retinal detachments (Figure 2a and Table 2) and one eye with a thickened posterior coat and suspicion of an intraocular foreign body (Figure 2b, arrow). These 13 eyes did not have surgery. The second eye of all these patients showed no posterior segment pathology, so unilateral cataract surgery was performed.

### 3.2. Ocular Adnexal Pathology

Patients with extraocular pathology were referred to a special clinic held in Saint Joseph’s Hospital Eye Department during visits by the Rostock group. There were 61 patients aged 6 months to 65 years over the period from 2006 to November 2019 examined both clinically and by B-scan ultrasound. In 20 of them, surgery was performed as a part of the orbital/adnexa sub-specialist service during the stay of the Rostock group at Saint Joseph’s Hospital. The surgeries included biopsies, tumor excision, mucocele drainage of the sinuses, and palliative treatment in extensive presumably malignant lesions. After diagnosis and evaluation, the remaining cases were referred to neurosurgical and maxillary facial departments for further management. Examples of all diagnoses are given in Table 4.

As an example, two cases with ocular adnexal pathology are demonstrated in Figure 2.

Example 1:

The patient presented with “conjunctival swelling”, which was treated with local antibiotics. Six months later, when the German surgical team arrived, he presented with a large mass pushing the lids apart (Figure 2a–c). A B-scan ultrasound displayed a normal posterior segment of the eye, and the oblique sections displayed intact iris structures (Figure 2b). Direct and indirect pupillary reactions were present in the eye behind the tumor. A diagnosis of a carcinoma of the conjunctiva was made on clinical grounds.

During surgery, a lid skin and orbicularis-saving extended enucleation was performed, described in detail elsewhere [8]. The patient recovered well and was still happy without sunglasses and no signs of local or obvious systemic recurrences 2 years postoperative (Figure 2c). Histopathology proved the suspected diagnoses [8].

Example 2:

A case of a 14-month-old child who presented with a 30-mm-diameter translucent orbital cyst in front of normally developed eyelids (Figure 2d–f). A B-scan ultrasound demonstrated a lesion involving the whole orbit, with no signs of ocular structures and no sound propagation into the surrounding sinuses (Figure 2e). There was no light perception on the affected side and no contralateral consensual pupillary reaction. During surgery, the cyst was excised totally, spearing the conjunctival lining. There was more than sufficient material to be attached to the upper and lower orbital rims with transcutaneous sutures to create new fornixes to host an artificial eye (Figure 2f).

A detailed histopathological examination of the excised material demonstrated rudimentary ocular structures, suggesting a final diagnosis of “microphthalmos with prominent cyst formation”, as described in detail elsewhere [9].

## 4. Discussion

### 4.1. Intraocular Pathology

All diagnostic procedures were discussed in Kinshasa in an open discussion of therapeutic consequences both with staff and patients. Additionally, the decision “not to treat” is kept as an important option. In our series, in a retrospective study of 437 eyes with blinding cataract, 13 eyes were excluded from cataract surgery due to retinal detachment diagnosed by B-scan ultrasonography, but with a full clinical diagnosis, only three of these patients would have had unnecessary surgery. Does this justify the expenditure (plus time and effort) for B-scan equipment? Our experience has also shown that ultrasound instrumentation needs special care and a budget for repair and replacement from time to time. Reviewing the literature of B-scan ultrasound applications in eyes with dense cataracts in low-income settings, the publications were mainly from India and Pakistan [10,11,12,13] and only a few from Sub-Saharan Africa [6,14]. In studies directly comparable with ours where the selected patients were screened, the rate of retinal detachment varied from 0.9% [13] to 4.5% [10], which was in line with our findings of 2.97%. In one study, the pathology was dominated by posterior staphyloma (3.5%) [10], but a similar pathology was only rarely documented by other authors (0.5%) [7].

The average frequency of retinal detachment in selected patients has been studied in detail by many authors [6,7,10,12,13,15]. Their studies have a total of 2854 patients with an average rate of retinal detachment of 3%. In contrast, Chanchlani & Chanchlani [11] found 0.94% retinal detachments in 400 eyes with hyper-mature cataract in India. Songrou and colleagues identified 21% retinal detachments in 46 posttraumatic eyes with opaque media, one-third complicated by vitreous hemorrhage [14].

Our results of 2.97% in 437 unselected eyes were in accordance with most of the other findings. There was only one prospective study [7] where the postsurgical findings were included in the overall consideration of 418 eyes, and due to the high-quality ultrasound equipment, even glaucomatous cupping of the disk played a role in the decision-making process for cataract extraction. The minor differences were in contrast to our study by using different instrumentation not available to us.

Some authors also correlated the clinical data with the presence of retinal detachment diagnosed by ultrasound, and these examinations identified unsuitable cases for cataract surgery [13]. Other findings were: posterior synechiae, keratic precipitates, elevated or reduced intraocular pressure, and an inaccurate light projection test [13]. This closely correlates with our results, where only three out of 437 eyes were diagnosed with retinal detachment without at least one of these clinical findings documented in the patients’ files. It is also probable that one of the indicators may have been overlooked in the daily work during a busy clinic. Additionally, in our study, eyes with posterior segment pathology were often accompanied by anterior segment changes (flat AC with low intraocular pressure (IOP) or neovascularization of the iris), so the indication for cataract surgery was already questionable. In this context, parasitic infections and keratitis should also be mentioned. For example, trachoma, one of the main causes of corneal opacities, is relatively rare in DR Congo. The more common onchocerciasis has been treated since the beginning of the 2000s with the distribution of ivermectin, so that there are only a few cases left in Kinshasa, which are mainly limited to diseases of the posterior segment, mostly around retinal degenerations (Ridley fundus). Additionally, Leprosy is now well under control and has thus become rare in Kinshasa. The same applies to Toxocara due to the low number of cats and dogs in Kinshasa, which means that more frequent outbreaks of these parasites and associated eye diseases are concentrated in the eastern part of the DRC and are therefore not part of our study.

What conclusion can be drawn about the management of cataract patients in low-income settings, where a B-scan ultrasound is rarely available and, if present, is related to extra costs for the patient or, at least, for the hospital? Salman et al. even found in their excellent prospective study that the clinical findings mentioned before were more strongly correlated to posterior segment pathology than a B-scan ultrasound itself [7]. We therefore strongly recommend a preselection of patients based on these clinical indications before performing or ordering any preoperative ultrasound evaluations. The likelihood of detecting a relevant pathology in eyes with advanced cataracts in patients without clinical signs is very low. Without careful preselection, extra unnecessary costs for all patients are inevitable. At least 80% of posterior segment pathology documented by a B-scan ultrasound could have been anticipated by interpreting standard clinical tests with more experience. As a result, in a department with approximately 1200 cataract extractions per year [16], where, in 600 eyes, there was no optical posterior segment evaluation possible, less than one patient would have been operated on without adequate visual improvement.

### 4.2. Ocular Adnexal Pathology

Before CT and MRI were available, ultrasonography played an important role in the diagnosis and planning treatment of ocular adnexal pathology. One of the crucial points in its application nowadays is the lack of experience in the use of ultrasounds by the clinicians and the availability of CT and MRI in specialized centers for orbital diseases. Furthermore, unlike CT and MRI, documented images in ultrasound examinations only recall snapshots of the dynamic examination procedure during which the diagnosis was made.

Despite the fact that CT is available in Kinshasa, the costs for an examination range between 150 and 300 USD, which nearly always exceeds the patients’ financial possibilities. The problem is worse for an MRI scan, the cost of this examination is even higher, ranging from $350 to 500, and MRI scanners are even more limited. These costs purely for imaging, which could certainly contribute to the best possible clinical outcome, are in addition to the necessary surgical costs to be paid by the patient and are not affordable for the majority of the population. Therefore, the authors’ approach was to investigate whether a diagnostic ultrasound can partly replace CT and MRI in the management of patients with orbital diseases in low-income settings. We also have to take into consideration that, with limited or no access to more costly imaging techniques, a B-scan ultrasound can be compared with cases some decades ago in Europe and the USA [17,18]. Nowadays, in industrialized countries, ultrasonography is nearly exclusively used for intraocular pathology, but it is still present in relevant comprehensive textbooks and even the selected newer literature for orbital pathology [19,20,21,22].

To answer the question of how helpful B-scan ultrasound imaging is in the management of orbital pathologies, one must consider that the overall incidence rate of orbital space-occupying lesions compared with all other diagnoses in ophthalmology is low [23]. In our series, we identified 32 pathological findings out of 61 patients with signs and symptoms of orbital pathology between 2006 and 2019. The most frequent diagnosis was orbital/lacrimal mucoceles, mainly posttraumatic. These patients were referred for surgery to external surgeons in Kinshasa, and some of them were operated on together with experienced ophthalmologists from Germany on-site [24]. In eight patients with enlarged eye muscles in conjunction with the typical clinical appearance, unilateral or bilateral dysthyroid eye disease was diagnosed. As there were no signs of optic nerve compression, all patients were managed conservatively [25].

Other diseases like av malformations were less frequent [26] and suggested in four patients, one of them with unilaterally raised intraocular pressure. Since there was no cranial trauma reported, a spontaneous Dural cavernous sinus fistula was suspected.

Equally rare were infiltrative lesions that turned out to be lymphomas, as proven by biopsy. One ill-defined lesion was found histologically to be an alveolar soft tissue sarcoma, which was treated surgically by orbital exenteration. In two other cases, one patient demonstrated an extensive carcinoma of the conjunctiva invading the anterior orbit (Figure 2a–c), and one other patient had a clinically diagnosed orbital cyst with nonfunctioning microphthalmos (Figure 2d–f), also shown by ultrasound examination. Both patients were successfully treated with surgery [8,9].

It must be mentioned again that, in a period of 13 years, we only saw 61 patients referred to the Rostock team in Kinshasa, a Sub-Saharan megacity with more than 10 million inhabitants. In about half of the patients, there were ultrasound findings that helped during management, either to make the clinical diagnosis more likely (e.g., Graves’ disease, venous, or av malformation) or to plan a surgery on the basis of a plain X-ray together with the typical ultrasound findings of mucoceles.

In the city of Kinshasa, the management of orbital patients is not centralized, but there has seemed to be a tendency towards referrals to Saint Joseph‘s Hospital over the years. Therefore, a teleconsulting platform has been established between Rostock and Kinshasa in order to assist in accurate diagnoses based on B-scan ultrasound imaging to support the development of the subspecialty of orbital diseases in the region and to further improve the clinical outcomes in the management of these rare diseases.

## 5. Conclusions

Summarizing the experience in a referral center in a third-world megacity with an under-resourced but nevertheless developing health system, a B-scan diagnostic ultrasound should be offered, and the application should be taught. Although the need for routine B-scan examinations in patients with dense cataracts referred for surgery seems to be limited and can most likely, in the authors’ experiences, be replaced by a thorough application of locally (freely) available examination techniques, such as the evaluation of a light projection test, IOP, pupillary reaction (RAPD), and careful slit lamp examination, its application is, however, recommended to manage patients with orbital pathology in the most cost-effective way. A B-scan ultrasound could provide the best possible clinical outcomes for these patients under the given circumstances, especially since the imaging modalities such as CT and MRI are severely limited in the region, and most of the population cannot afford these scans.

## Figures and Tables

**Figure 1 diagnostics-11-02009-f001:**
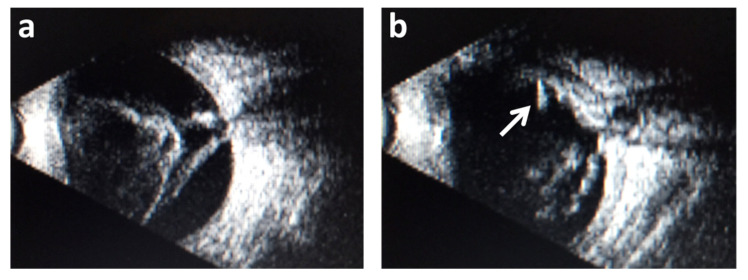
Findings of the B-scan ultrasonic examination. (**a**) Retinal detachment and (**b**) an intraocular foreign body (white arrow). Posterior segment abnormalities were most commonly associated with posterior uveitis, followed by undetermined and posttraumatic cataracts. Other causes had relatively low incidences of posterior segment lesions, respectively (Table 2).

**Figure 2 diagnostics-11-02009-f002:**
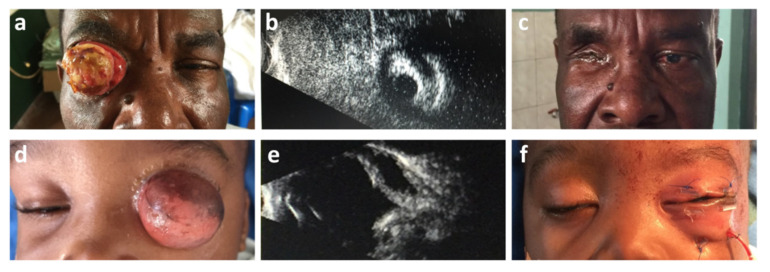
Ocular adnexal pathology. (**a**) Large carcinoma of the conjunctiva, tumor not appearing to be invading the lid margins. (**b**) B-scan ultrasound displaying a solid mass in front of the anterior segment of the eye. Sagittal section through the anterior segment of the eye displays an iris structure with a light-sensitive pupil. (**c**) Postoperative results 16 months after lid skin-saving extended enucleation and a total permanent tarsorrhaphy. (**d**) Fourteen-month-old child with a unilateral translucent orbital cyst. (**e**) B-scan ultrasound proves the cystic character of the lesions, which reaches the orbital walls up to the apex. (**f**) Postsurgical situation after excision and wound closure with transcutaneous fornix deepening sutures fixed over the bolsters.

**Table 1 diagnostics-11-02009-t001:** B-scan ultrasonography findings.

Posterior Segment Pathologies	Frequencies (*n*)	Proportions (%)
Posterior Vitreous Detachment	15	3.43
Retinal Detachment	13	2.97
Ill-defined signals from vitreous cavity	10	2.29
Disseminated high reflective mobile structures (after triamcinolone injection)	1	0.23
No ultrasound signals from the vitreous cavity	398	91.08

**Table 2 diagnostics-11-02009-t002:** Posterior Segment pathology variations according to the etiologies of cataracts.

Types of Cataracts	Frequencies (*n*)	Posterior Vitreous Detachment	Retinal Detachment	Ill-Defined Signals from Vitreous Cavity	Total	Proportion (%)
Congenital	92	1	0	0	1	1.09
Undetermined	4	3	0	0	3	75
Juvenile	3	0	0	0	0	0
Diabetic	36	1	2	3	6	16.67
Age-related	284	9	5	4	18	6.34
Posttraumatic	10	1	2	1	4	40
Post-Uveitic	8	0	4	3	7	87.50
**Total**	**437**	**15**	**13**	**11**	**39**	8.92

**Table 3 diagnostics-11-02009-t003:** Clinical risk factors for retinal detachment (*n* = 13).

Clinical Parameters	Frequencies (*n*)	Proportions (%)
Posterior synechiae	8	61.54
Ocular hypotony (≤5 mmHg)	4	30.77
Inaccurate light projection test	10	76.92
None	3	-

**Table 4 diagnostics-11-02009-t004:** Diagnoses of ocular adnexal space occupying lesions by ultrasound seen in Saint Joseph’s Hospital Eye Department between 2006 and 2019.

Diagnosis	Frequencies (*n*)
Orbital/lacrimal mucoceles (mainly posttraumatic)	9
Dysthyroid eye diseases (myogenic exophthalmos)	8
av malformations	4
Orbital lymphomas (biopsy proven)	4
Ill-defined infiltrative orbital lesions (see Figure 2a–c)	4
Orbital involvement of presumed sphenoid wing meningioma (with CT-scans in patients’ hands)	2
Anophthalmus with cyst (see Figure 2d–f)	1
No pathological findings on B-scan ultrasound	29

## Data Availability

The datasets used and/or analyzed during the current study are available from the corresponding author upon reasonable request.

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
