# Peer review of "Ophthalmic Ultrasonography in Sub-Saharan Africa—A Kinshasa Experience"

_diagnostics, 2021, doi:10.3390/diagnostics11112009_

Round 1
Reviewer 1 Report
Dear Authors,
thank You for interesting manuscript analyzing the usefulness of the diagnostic B-scan ultrasound in low and middle income countries. In the retrospective study twaogroups are analyzed: posterior segment of the eye in dense cataract patients and (group 2) extraocular pathology.
The description of groups and results are clear, however it is not clear what authors understand under the term of "diabetic cataract", also uveitis group could be desctibed more precisely. It could be also valuable to check statistical correlation between pathologies and groups.
As the main conlusion is the matter of costs, this issue should be more discussed.
in line 19 please use other construction (it looks like 261 patients).
I tecoomend the manuscript for publication after major revision.
Friendly regards.
Reviewer 2 Report
This is an interesting paper which, in my opinion, would be enriched if the authors consider the following:
i) The authors concentrate more on what they found (ie the numbers of presenting conditions) and little on the ‘risk-benefit’ of investing in more expensive diagnostic procedures. They should show provide (eg in a table) a breakdown of the extra cost/patient (including cost of extra space, duration of visit and extra specialist) if the envisaged extra diagnostic procedures were implemented. This should include tele-consultations. Why? Because of the global shift towards more remote/online consultation in all aspects of health care brought about by Covid-19.
ii) Please refrain from using outmoded terms that many find insulting (eg senile). Replace with more acceptable terms.
iii) I was surprised the authors found no cases of either current or past intra-ocular parasitic activity or mention of Keratitis per se. Please comment.
iv) The authors wrote ‘The likelihood of detecting relevant pathology in eyes with advanced cataract in patients without clinical signs is very low.’ Include the numbers and, by extrapolation, predict the likely numbers that will have ‘relevant pathology in eyes with advanced cataract’ for the territory over, say, 1 year.
Round 2
Reviewer 2 Report
The revision is an improvement of the original manuscript. However, the authors’ comments regarding keratitis, onchocerciasis and leprosy should be summarised and included in the Discussion. Why? Because this would interest readers and scholars alike.
Author Response
Reviewer’s comment #1: |
The revision is an improvement of the original manuscript. However, the authors’ comments regarding keratitis, onchocerciasis and leprosy should be summarised and included in the Discussion. Why? Because this would interest readers and scholars alike. |
|
|
Authors’ response:
|
Thank you very much for this assessment. In line with the reviewer's recommendation, we have included a short paragraph on keratitis and parasitic infections in the discussion.
|
Changes in manuscript: |
Line 245-253: In this context, parasitic infections and keratitis should also be mentioned. For example, trachoma, one of the main causes of corneal opacities, is relatively rare in DR Congo. The more common onchocerciasis has been treated since the beginning of the 2000s with the distribution of ivermectin, so that there are only a few cases left in Kinshasa, which are mainly limited to diseases of the posterior segment, mostly around retinal degenerations (Ridley fundus). Also Leprosy is now well under control and has thus become rare in Kinshasa. The same applies to Toxocara due to the low number of cats and dogs in Kinshasa, which means that more frequent outbreaks of these parasites and associated eye diseases are concentrated in the eastern part of the DRC and are therefore not part of our study. |
